# Epibiotic Fungal Communities of Three *Tomicus* spp. Infesting Pines in Southwestern China

**DOI:** 10.3390/microorganisms8010015

**Published:** 2019-12-20

**Authors:** Hui-Min Wang, Fu Liu, Su-Fang Zhang, Xiang-Bo Kong, Quan Lu, Zhen Zhang

**Affiliations:** The Key Laboratory of Forest Protection, Research Institute of Forest Ecology, Environment and Protection, Chinese Academy of Forestry, National Forestry and Grassland Administration, Beijing 100091, China; wanghuimin@caf.ac.cn (H.-M.W.); liufu2006@163.com (F.L.); Zhangsf@caf.ac.cn (S.-F.Z.); xbkong@sina.com (X.-B.K.)

**Keywords:** community structure, high-throughput sequencing, ITS1, *Tomicus yunnanensis*, *T. minor*, *T. brevipilosus*

## Abstract

The association between insects and fungi has evolved over millions of years and is ubiquitous in nature. This symbiotic relationship holds critical implications for both partners, the insects and the associated microbes. Numerous fungi are externally allied with bark beetles and form a close symbiosis, but the community structures of these fungi are largely unknown. In Yunnan Province in southwestern China, the beetles *Tomicus yunnanensis*, *T*. *minor*, and *T*. *brevipilosus* are major forest pests that cause large losses of two indigenous pines, *Pinus yunnanensis* and *P*. *kesiya*. In this study, we used the Illumina MiSeq PE300 platform to process 48 samples of epibiotic fungal communities pooled from 1348 beetles; the beetles were collected during both the branch- and trunk-infection sections from five locations across Yunnan Province. Considerably greater species richness was detected using high-throughput sequencing of amplified internal transcribed spacer 1 (ITS1) ribosomal DNA (rDNA) libraries than previously documented by using culture-dependent methods. In total, 1,413,600 reads were generated, and a 97% sequence-similarity cutoff produced eight phyla, 31 classes, 83 orders, 181 families, 331 genera, 471 species, and 1157 operational taxonomic units (OTUs), with 659, 621, and 609 OTUs being confined to *T*. *yunnanensis*, *T*. *minor*, and *T*. *brevipilosus*, respectively. *Tomicus yunnanensis*, *T*. *minor*, and *T*. *brevipilosus* had the similar OTUs richness and evenness of fungal communities in Yunnan Province; nevertheless, the two fungal community compositions associated with *T*. *yunnanensis* and *T*. *minor* were structurally similar to each other but distinct from that associated with *T*. *brevipilosus*. Lastly, the results of principal co-ordinates analysis suggested that epibiotic fungal community structures of the three *Tomicus* spp. were conditioned strongly by the locations and pine hosts but weakly by beetle species and infection sections. Our findings provide baseline knowledge regarding the epibiotic fungal communities of three major *Tomicus* spp. in southwestern China.

## 1. Introduction

The association between forest insects and microorganisms has been documented as a “key driver” of biotic damage to trees worldwide [1,2,3,4,5]. For example, bark beetles (Coleoptera: Curculionidae: Scolytinae), which constitute a group of the most successful herbivorous insects, are associated with diverse microorganisms [6,7]. Von Schrenk (1903) first reported >100 years ago that trees killed by bark beetles were frequently contaminated, within a few weeks of the attack by fungi, which were potentially transmitted by the beetles [8]. The bark beetle-fungus interactions benefit both groups of these organisms; for example, symbiotic fungi can provide essential nutrients to their vectors [9], detoxify tree defenses [10], and enhance beetle population aggregation to enable successful infection of host trees [11,12,13], whereas the fungi can be inoculated into new host plants through the phoresy and invasion of the bark beetles [13]. Bark beetle species are particularly associated with ophiostomatoid fungi, which can cause extensive damage to *Pinus* spp. or other forest trees worldwide [2,5,14].

Most bark beetle species infest *Pinus* spp. [15,16], with several of these beetles being regarded as secondary intruders that colonize dead, diseased, or weakened trees [17,18]; however, a few beetles, such as those of *Dendroctonus valens* LeConte, *Ips typographus* Linnaeus, *Scolytus multistriatus* Marsham, and *Tomicus yunnanensis* Kirkendall and Faccoli, can colonize and kill healthy hosts [19,20]. *Tomicus* spp., which are among the most destructive beetles, can kill large numbers of healthy trees during periodic outbreaks [21,22]. In Yunnan Province in southwestern China, two indigenous pines, *Pinus yunnanensis* and *P*. *kesiya* are the most important afforestation coniferous trees and allopatrically distribute in north and south of Yunnan Province, respectively. They played invaluable roles in the local ecology and economy. They are the primary hosts of *Tomicus yunnanensis*, *T*. *minor*, and *T*. *brevipilosus*. The three pine-shoot beetles share an overlapping geographical-distribution range and infection periods; two or three of these beetle species frequently coexist underneath the barks or shoots of a single host [21,22]; the three beetles complete their life cycles entirely inside the pines at either the stem section or branch section of infestation [23,24]. These beetles aggregately infect both pines [24,25,26,27,28], and the beetles have caused pine-wood losses totaling >93,000 m^3^ in southwestern China since the 1980s [22,29,30]. 

Most fungal-biodiversity studies on beetle ectosymbioses have been conducted using culture-dependent methods and traditional molecular techniques to identify the fungi obtained in pure cultures. However, the traditional approaches of analyzing fungal communities are limited in throughput and affected by methodological biases of culture-based techniques, which could lead to biases in fungal-diversity characterization due to the low number of isolates recovered. By comparison, the application of next-generation sequencing technologies, such as high-throughput sequencing (HTS), can overcome these limitations and thus help reveal a higher taxonomic diversity and richer fungal-community structure associated with bark beetles [6,14,31,32,33].

In this study, we used the ITS1 rDNA Illumina MiSeq PE300 platform to perform the first high-throughput survey of epibiotic fungi associated with *T*. *yunnanensis*, *T*. *minor*, and *T*. *brevipilosus*. We aimed to unravel the fungal diversity and community composition of the three *Tomicus* beetles. We determined whether the fungal-community structures were specific to beetle species, infestation sections, pine-host species, or geographic locality, and we also verified whether the results of a culture-independent approach would corroborate the taxonomic identities of fungi associated with *Tomicus* spp. that were previously determined based on a culture-dependent study [34].

## 2. Materials and Methods 

### 2.1. Sample Collection

Mature adults of bark beetles at the branch- and trunk-infection sections were used for analyzing extraneous mycobiota. Samples of the three pine-shoot beetles were collected from *P. yunnanensis* and *P*. *kesiya* at five distinct locations in Yunnan Province in December 2016 and January–March 2017 (Appendix A): Anning, Yuxi, and Qujing sampling sites are located around the provincial capital Kunming, in eastern Yunnan Province, belonging to the Yunnan-Guizhou Plateau, whereas Xiangyun and Pu’er sampling sites are far away from Kunming city and located in western Yunnan Province. The Xiangyun sampling site is located in southward extension of the Hengduan Mountains. Pu’er in southwestern Yunnan Province is afforested by *P*. *kesiya* and other four locations are by *P. yunnanensis*; therefore, the beetle samples are generated from these two different pines. The bark beetle samples were collected from trunk phloems of the moribund hosts and withered branches of the green hosts. Bark beetles were transferred into 2 mL Eppendorf tubes individually, placed in envelopes, and stored at −80 °C until analysis. In total, 1348 shoot beetles were collected and divided into 48 samples for further metagenomic analysis (details in Appendix A). Due to the low concentration of epibiotic fungal DNA of a single beetle, 25–30 beetles were pooled as a sample, depending on the pre-experimental results and number of beetles collected. The bark beetles of the trunk-infection section in a single sample were selected from the single or adjacent galleries of a single infesting pine, and the beetles of branch-infection section in a single sample were selected from the adjacent branches of a single infesting pine. Bark beetles in each sample were therefore regarded as homogeneous and were pooled together.

### 2.2. DNA Extraction, PCR Amplification, and Sequencing

Before extracting epibiotic fungal DNA, all beetles from each sample were surface-washed concurrently by vortexing for 15 s in sterile phosphate-buffered saline; after washing thrice, the ecto-associated fungi of the bark beetles were collected from the suspension through a 0.2 μm filter membrane.

Fungal DNA was extracted using a E.Z.N.A.^®^ soil DNA Kit (Omega Bio-tek, Norcross, GA, USA), according to the manufacturer’s protocol. The final concentration and purity of DNA were determined using a NanoDrop 2000 UV-vis spectrophotometer (Thermo Scientific, Wilmington, DE, USA). The marker used for the fungal-community survey was ITS1 of nuclear rDNA, on average 300 bp long, and the region was amplified using the primers ITS1F (5’-CTTGGTCATTTAGAGGAAGTAA-3’) and ITS2R (5’-GCTGCGTTCTTCATCGATGC-3’) [35] and a thermocycler PCR system (GeneAmp 9700, ABI, Waltham, MA, USA).

PCR was conducted using the following program: 3 min of denaturation at 95 °C; 35 cycles of 30 s at 95 °C, 30 s of annealing at 55 °C, and 45 s of elongation at 72 °C; and a final extension at 72 °C for 10 min. All reactions were performed in triplicate 20 μL mixtures containing 4 μL of 5 × FastPfu Buffer, 2 μL of 2.5 mM dNTPs, 0.8 μL of each primer (5 μM), 0.4 μL of FastPfu Polymerase, and 10 ng of template DNA. The generated PCR products were extracted from 2% agarose gels and further purified using an AxyPrep DNA Gel Extraction Kit (Axygen Biosciences, Union City, CA, USA) and quantified using QuantiFluor™-ST (Promega, Madison, WI, USA), according to the manufacturer’s protocol.

### 2.3. Library Construction

Purified amplicons were pooled in equimolar amounts and paired-end sequenced at 2 × 300 bp by using PE300 strategies on an Illumina MiSeq platform (Illumina, San Diego, CA, USA) according to standard protocols by Majorbio Bio-Pharm Technology Co. Ltd. (Shanghai, China). The raw reads were deposited into the NCBI Sequence Read Archive (SRA) database (Accession Number: SRP173175).

Raw Illumina fastq files from 48 samples were demultiplexed, quality-filtered using Trimmomatic, and analyzed using FLASH, with the following criteria: Reads were truncated at any site that received an average quality score of <20 over a 50-bp sliding window; primers were exactly matched, allowing for 2 nt mismatch, and reads containing ≥1 ambiguous bases were removed; and sequences whose overlap was >10 bp were merged according to their overlap sequence.

ITS1 reads were clustered into operational taxonomic units (OTUs) by using UPARSE [36] with a cutoff threshold of 97% similarity to reference sequences, and chimeric sequences were identified and removed using UCHIME. For each OTU, the most abundant read was designated as the representative sequence, and all representative sequences were aligned against Unite database and identified taxonomically (at phylum, class, order, family, genus, and species levels). To compare the plateau and control samples at the same sequencing level, the minimum read number was used to subsample sequences from all other samples, and then OTUs were identified taxonomically by RDP Classifier algorithm [37] against Unite database [38] by using a confidence threshold of 70%.

### 2.4. Data Analysis

Based on the OTU information, the abundance and uniformity of fungal taxon samplings were statistically analyzed by generating rarefaction curves. Samples were homogenized with respect to the sample featuring the lowest number of reads. For the fungal communities in each of the three beetle species, α-diversity was estimated to determine community richness (Chao1), community diversity (Shannon), and sequencing depth (Good’s coverage) [39]. Student’s *t* test was used to compare fungus diversity.

By using the data on fungal OTU presence/absence, Venn diagrams were generated, using the web application Venny [40], for the following comparisons among the total fungal communities of the three *Tomicus* spp. The diagrams of community-composition analysis and heatmap were used to compare fungal-community composition; OTUs whose abundance was <0.01% were combined into other OTUs in the diagrams of community-composition analysis.

Sample similarities were visualized using principal co-ordinates analysis (β-diversity). For all categorical classifications of beetle species, hosts, infectious sections and locations, their interactions with the fungal composition of samples were tested using a complementary nonparametric approach: analysis of similarities [41] (ANOSIM). One-way ANOVA (analysis of variance) and Wilcoxon rank-sum tests were used to identify differences in community richness and evenness. 

## 3. Results

### 3.1. Data Characteristics

In this study, 1,413,600 reads were generated after quality filtering, denoising, and chimera removal based on the metagenomic analysis of 48 samples. After demultiplexing each sample (i.e., after assigning reads to beetle samples) to an equal sequencing depth, the average read count per sample was 29,450. The 97% sequence-similarity cutoff yielded eight phyla, 31 classes, 83 orders, 181 families, 331 genera, 471 species, and 1157 clusters representing OTUs featuring at least two reads. The Good’s coverage for the observed OTUs was 99.95. The gradually flattening rarefaction curves of OTU richness (Appendix A) indicated that the vast majority of the fungal community was captured in all samples.

### 3.2. Epibiotic Community Diversity

Among the three pine-shoot beetle samples, the Chao1 estimate of total diversity in the case of each beetle species ranged from 101.38 to 139.9, whereas the Shannon diversity index of each species ranged from 1.62 to 1.75 (Table 1). The diversity and richness were lower for *T. brevipilosus* samples than for *T*. *yunnanensis* and *T*. *minor* samples at the branch infection section but significant differences were not detected among three beetle species. Comparison of the two infectious sections further revealed that the fungus diversities of *T*. *yunnanensis* and *T*. *minor* samples were higher in the branch-infection than trunk-infection section, but the opposite was recorded with *T*. *brevipilosus* samples (Table 1).

Venn-diagram plotting showed that 659, 621, and 609 OTUs were confined to each of the three beetle species, with 217 OTUs being common to all groups (Figure 1a). Moreover, the fungal community of *T*. *yunnanensis* was similar to that of *T*. *minor*, with 401/879 OTUs being shared here, which was considerably more than the 263/1005 and 285/945 OTUs shared by the *T*. *yunnanensis* and *T*. *brevipilosus* groups, the *T*. *minor* and *T*. *brevipilosus* groups, respectively. Examination of microbial-community composition also showed that the associated fungal communities of three pine-shoot beetle samples were similar to each other in their richness, evenness, but that the fungal community compositions were markedly different (Figure 1b). The fungal communities of *T*. *brevipilosus* were heavily dominated by a single OTU 406 (Saccharomycetales: Unclassified-genus), which accounted for 46.3%, while *T*. *yunnanensis* and *T*. *minor* were heavily dominated by OTU476 (Saccharomycetales: *Yamadazyma mexicana*), which accounted for 35.5% and 51.24%, respectively. Lastly, when a heatmap was used to visualize the relative abundance of the major genera (top 50) according to beetle species, about one-third of the significant genera were found to score zero in the case of *T*. *brevipilosus* relative to those that were abundant in *T*. *yunnanensis* and *T*. *minor* (Figure 1c).

### 3.3. Taxonomic Identities of Fungal Communities 

Taxonomic analysis of fungal sequences, obtained using the Illumina MiSeq PE300 platform, in the Unite reference database revealed that the phyla in the fungal communities associated with *T*. *yunnanensis*, *T*. *minor*, and *T*. *brevipilosus* were Ascomycota, Basidiomycota, Glomeromycota, Blastocladiomycota, Chytridiomycota, Mortierellomycota, Mucoromycota, and Rozellomycota. Of these, Ascomycota featured the highest number of reads, 1,372,776 (97.1%), and this was followed by Basidiomycota, with 30,385 reads (2.1%).

We identified eight orders featuring >10,000 reads: Capnodiales, Eurotiales, Hypocreales, Ophiostomatales, Pleosporales, Saccharomycetales, Sordariales and unclassified_p_Ascomycota (Appendix A). The three orders with the majority of reads were Saccharomycetales (939,262 reads, 66.44%), Ophiostomatales (102,167 reads, 7.23%), Eurotiales (97,547 reads, 6.9%), and the remaining five orders featured the minority of the reads (195,261 reads, 13.81%) (Appendix A). Excluding unclassified genera, Saccharomycetales, Ophiostomatales and Eurotiales included eleven, five, eight genera, respectively (Appendix A). The top 10 genera of the fungal communities associated with the three *Tomicus* spp. were composed of six genera from the three predominant orders (Appendix A) (*Yamadazyma*, *Ophiostoma*, *Penicillium*, *Nakazawaea*, *Talaromyces*, and unclassified _o_Saccharomycetales) and four genera from the remaining five orders (*Alternaria*, *Lapidomyces*, unclassified_p_Ascomycota, and unclassified_o_Hypocreales).

### 3.4. Factors Influencing Fungal Communities and Abundance Among P. yunnanensis Samples

The β-diversity of the fungal OTUs in the three communities at 97% sequence similarity was examined using principal co-ordinates analysis. A best-variables rank-correlation test (BEST) was used to evaluate the factors that explain the β-diversity dissimilarity degree of the fungal structures, after which permutational MANOVA tests were used to confirm significance. Samples of the three pine-shoot beetles were collected from *P. yunnanensis,* and only *T*. *brevipilosus* samples were also collected from *P*. *kesiya* at Pu’er in southwestern Yunnan Province (Appendix A). So, we chose samples collected from *P. yunnanensis* to explain the β-diversity dissimilarity degree of fungal-community structures by beetle species, infectious section, and location. 

The generated plots showed that fungal communities were significantly or weakly affected by beetle species, infectious section, and location (Figure 2, Figure 3 and Figure 4). Repetitive samples showed little differences in the fungal community, which indicated that the results were reliable. 

#### 3.4.1. Beetle Species

Fungal-community structures were slightly influenced by beetle species. Weak separation was observed between *T*. *yunnanensis*, *T*. *minor* and *T*. *brevipilosus* (Figure 2a). The results of MANOVA showed that beetle species was an insignificantly influencing factor related to fungal patterns in all samples (R_ANOSIM_ = 0.25, *p* = 0.002; Figure 2a). Among the top 10 genera of the fungal communities associated with the three *Tomicus* spp. collected from *P. yunnanensis*, two genera (unclassified-o-Saccharomycetales, and *Yamadazyma*) differed significantly in abundance (*p* < 0.05) but the remaining eight genera did not (Figure 2b).

#### 3.4.2. Infectious Section

The scale and impact of infectious sections on fungal-community structure were elucidated using two infectious sections, the branch- and trunk-infection sections of *P. yunnanensis* (Figure 3a). Infectious section only weakly or insignificantly affected overall fungal-community structures (R_ANOSIM_ = 0.12, *p* < 0.005; Figure 3a). Among the top 10 genera of the fungal communities associated with the two infectious sections, four genera (*Kuraishia*, *Ophiostoma*, *Pestalotiopsis*, and unclassified-o-Saccharomycetales) were significantly different in terms of abundance (*p* < 0.05) and the remaining six were not (Figure 3b).

#### 3.4.3. Location

Sampling points played a major role in fungal-community structures and clustering patterns across the eastern and western province (Figure 4a,c). Location strongly affected fungal-community structures: three sites, Anning, Yuxi, and Qujing near each other, are located around provincial capital Kunming, in eastern Yunnan Province, whereas the other site is located at a distance: Xiangyun, in western Yunnan Province. MANOVA results showed that different sampling points were significant influencers related to fungal patterns in all samples (*T*. *yunnanensis*: R_ANOSIM_ = 0.71, *p* = 0.001; *T*. *minor*: R_ANOSIM_ = 0.79, *p* < 0.002; Figure 4a,c). Among the top 10 genera of the fungal communities associated with the eastern/western of *T*. *yunnanensi* and *T*. *minor*, five genera (*Nakazawaea*, *Penicillium*, unclassified-f-Hypocreales, unclassified-f-Chaetomiaceae, and *Yamadazyma*) were significantly different in terms of abundance (*p* < 0.05) and the remaining five were not (Figure 4b) by *T*. *yunnanensis* samples; six genera (*Debaryomyces*, *Lapidomyces*, *Neocamarosporium*, unclassified-f-Hypocreales, unclassified-f-Chaetomiaceae, and *Yamadazyma*) were significantly different in terms of abundance (*p* < 0.05) and the remaining four were not (Figure 4d) by *T*. *minor* samples.

### 3.5. Hosts Influencing Fungal Communities and Abundance Associated with T. brevipilosus 

All *T*. *brevipilosus* samples collected from *P. yunnanensis* and *P*. *kesiya* were used to explain the β-diversity dissimilarity degree of fungal-community structures by host.

A high degree of correlation was observed between different hosts (*P*. *yunnanensis* in Qujing, northeast of the province and *P*. *kesiya* in Pu’er, southwest of the province) and fungal clustering patterns (Figure 5a). *Pinus yunnanensis* and *P*. *kesiya* were distributed in different topography in Yunnan Province, so fungal-community structures appeared to be strongly influenced by host and topography (location). MANOVA results identified host as a significant influencer related to fungal patterns in *T*. *brevipilosus* samples (R_ANOSIM_ = 1, *p* = 0.007). Among the top 10 genera of the fungal communities associated with the two hosts, six genera (*Cladosporium*, *Geosmithia*, *Lapidomyces*, *Rachicladosporium*, unclassified-p-Ascomycota and unclassified-o-Saccharomycetales) differed significantly in abundance (*p* < 0.05) but the remaining four did not (Figure 5b).

## 4. Discussion

This is the first study in which the HTS metabarcoding approach was used to explore the fungal-community structures, including composition and diversity, associated with three pine-shoot beetles (*T*. *yunnanensis*, *T*. *minor*, and *T*. *brevipilosus*) that concomitantly infect *P*. *yunnanensis* and *P*. *kesiya*. The HTS method has been previously used to analyze the associated fungal-community structure of several bark beetles, such as *Dendroctonus* beetles (*D*. *micans*, *D*. *punctatus*, and *D*. *valens*), *Hylastes ater*, *Orthotomicus erosus*, *Tomicus piniperda*, *Trypodendron lineatum*, and ambrosia beetles (*Xyleborus affinis*, *Xyleborus ferrugineus*, *Xyleborinus saxesenii*, and *Xylosandrus crassiusculus*) [6,14,32,33]. Miller et al. [33] reported that 435 OTUs were obtained from *T*. *piniperda*. Because extensive data filtering can potentially skew species composition, OTU numbers between this study and previously studies cannot be readily compared, but certain types of fungal-community composition were similar; for example, yeast and ophiostomatoid fungi were abundant and frequent among bark beetle species [14,31,33].

Species richness and evenness of fungal communities were similar between *T*. *yunnanensis*, *T*. *minor*, and *T*. *brevipilosus* (Table 1 and Figure 1). *Tomicus yunnanensis* and *T*. *minor* featured similar fungal-community structures that were distinct from those of *T*. *brevipilosus* (Figure 1b,c). We speculate that the drivers responsible for this in the case of *T*. *brevipilosus* could be the following: samples of *T*. *brevipilosus* were collected from *P. yunnanensis* and *P*. *kesiya*, while samples of *T*. *yunnanensis* and *T*. *minor* were only collected from *P. yunnanensis*. Other underlying factors could be the divergent reproductive biology and patterns of infection of different beetles. The breeding attack timing of *T*. *brevipilosus* is obviously later than those of other two beetles, with two infection patterns (following the attacks by *T. yunnanensis* and *T. minor* in same host, or attacking the host by itself) [23,24]. The period hosts were infected and their healthy status could influence the fungal communities associated with bark beetles [42]. So, the later breeding attack and the different infection patterns comparing to *T*. *yunnanennsis* and *T*. *minor* could lead to the difference of the fungal community associated with *T*. *brevipilosus*. Bark beetles carry, according to their requirements, fungi showing different levels of diversity and fungi of distinct types.

In this study, beetle species, infectious sections, locations, and host were identified as predictors of fungal-community structures (Figure 2, Figure 3, Figure 4 and Figure 5) based on both nonparametric tests and principal co-ordinates analysis. The effects were small in beetle species and infectious sections but generally stronger in locations and hosts, which accounted for 71% and79% of the variation among *T*. *yunnanensis* samples and among *T*. *minor* samples collected from *P*. *yunnanensis*, respectively and 100% of the variation among *T*. *brevipilosus* from *P. yunnanensis* and *P*. *kesiya* samples. The study of Skelton et al. [43] considered that fidelity between phloem-feeding bark beetles and associated fungi is even less settled, similar to our results: beetle species only weakly affected overall fungal-community structures. Meanwhile, he though the fungal communities in mycangia of beetles might be more stable than epibiotic fungal communities. Just like the research of Kostovcik et al. [14], the fungal communities in mycangia of three ambrosia beetles were predicted by beetle species and location. They suggested that the fungal communities were predicted considerably more strongly by beetle species than by location; in contrast to these results, the epibiotic fungal communities of the three *Tomicus* spp. here were predicted markedly more strongly by location than by beetle species. Zhou et al. [44] reported that the fungi associated with *T*. *yunnanensis*, on which mycangia were not observed, they were evenly distributed on the body surface. Thus, we infer that fungal communities of pine-shoot beetles or other bark beetles without mycangia might significantly be influenced by location, host, or other factors except for the beetle species, especially under similar habits and close environments. Furthermore, our results similar to the study of Dohet et al. [6], who reported, respectively, that three *Dendroctonus* bark beetle species showed similar fungal communities despite inhabiting distinct geographic regions. The beetles studied by Dohet et al. [6] were cultured in the laboratory for several generations, and fungal communities might be biased in the beetles directly collected from the field. 

The topography of Yunnan Province is complicated. The five sampling sites could be generally categorized into three types: eastern Yunnan (Anning, Yuxi, and Qujing), northwestern Yunnan (Xiangyun) and southwestern Yunnan (Pu’er). To be interesting, the fungal communities correspondingly show three distinct patterns (Figure 4 and Figure 5) no matter with which kind of beetles associate, evidently indicating that location and host strongly influence the fungal-community structures.

In addition to the four aforementioned factors, differences in infesting intensity of beetles and sample heterogeneity might influence fungal diversity and abundance. The infesting intensity of the three bark beetles differs: *T. yunnanensis* is the most aggressive of the three species, being capable of primarily attacking and killing healthy *P*. *yunnanensis* trees [21,23,25,45]; *T. minor*, which typically infests host trees that are already attacked by *T*. *yunnanensis* or *T*. *brevipilosus*, is frequently regarded as an opportunist that facilitates tree deaths [16,26,46]; and *T*. *brevipilosus* shows two patterns, either colonizing the trunks of host trees already infested by both *T*. *yunnanensis* and *T*. *minor*, or primarily infecting healthy trees and thus exhibiting higher aggressiveness than *T*. *minor* [23,25]. The virulence of the fungi associated with invasive beetles was reported to be considerably stronger than that of fungi associated with secondary beetles [10], which imply that the fungi associated with distinct beetles are different. Therefore, the difference in infesting intensity of the three bark beetles influences the communities and abundance of associated fungi. Another point to consider is that sampling time and sampling trees were artificially defined, and thus the recorded effects would likely not be entirely completely consistent; sample heterogeneity influences fungal communities and abundance. In summary, marked differences were observed in high-abundance fungi (Figure 1c), and the abundance of several associated fungi differed substantially, which indicates that the fungi carried by bark beetles are distinct and that bark beetles likely carry some neutral or beneficial fungi preferentially under certain conditions, depending on their own requirements.

Ophiostomatoid fungi are probably the most intensively and accurately investigated fungi associated with the bark beetles [43,45,47,48]. Several ophiostomatoid fungi associated with the three *Tomicus* spp. have been classified as the dominant mutualistic associates. We collected a series of the primary literature and extracted records of ophiostomatoid fungi associated with the three *Tomicus* spp. in Yunnan Province (Table 2); 16 species belonging to five genera (*Ophiostoma*, *Sporothrix*, *Leptographium*, *Graphilbum*, and *Esteya*) in Ophiostomatales were documented based on culture-dependent studies, and, interestingly, five genera in Ophiostomatales were detected through HTS analysis in this study (Table 3).

Certain differences in the genera in Ophiostomatales were observed between the culture-dependent studies and culture-independent analysis (Table 2 and Table 3). For example, a representative of the genus *Ceratocystiopsis* was detected in this study but not obtained from culture-dependent studies. Although only one strain of the fungus was found, *Esteya vermicola* was obtained in culture-dependent studies but was not detected in the HTS analysis; *E. vermicola* is widely distributed worldwide [49,50,51,52], but invariably only one strain is obtained. We speculate that because of the limited number of reads generated, the HTS analysis was unable to help with precise detection of *Esteya*. Furthermore, only one *Grosmannia* species, *G. yunnanense*, was previously reported to be associated with *Tomicus* spp. [43,45,53], but our HTS analysis yielded two OTUs of *Grosmannia* representing potentially two species (*G. aurea* and *G*. *yunnanense*). Moreover, the number of reads and the OTUs of *Ophiostoma* were abundant and frequent in *Tomicus* spp., and the dominance of *Ophiostoma* in this study was similar to that found in culture-dependent studies: *O. brevipilosi*, *O*. *canum*, and *O*. *minus* were the predominant fungi associated with *Tomicus* spp. [34]. The most abundant reads were potentially representative of two species, *O*. *brevipilosi* and *O*. *canum*, which were the preponderant ophiostomatoid fungi from *T*. *brevipilosus* and *T*. *minor*, respectively. These findings suggest that the diversity estimates in this study likely approach the genuine diversity, and that our sampling was nearly exhaustive. Overall, our results showed that the diversity of the fungi associated with the three *Tomicus* spp. in this study was an order of magnitude higher than that previously reported from culture-dependent studies [43,45,47,48,54].

Beetle species was an influencing factor related to fungal-community structures in this study (Figure 2a), nevertheless, the association of three *Tomicus* spp. and some ophiostomatoid fungi are preferential even species-specific association in the pine forest of Yunnan Province based on culture-dependent studies and culture-independent analysis (Table 2 and Table 3). This may mean that there are some symbioses between ophiostomatoid fungi and *Tomicus* spp., but it is difficult to identified symbioses under too many incidental associations. 

In addition to ophiostomatoid fungi, several other fungal genera were frequently detected in this study, however, few studies to date have reported the association of these genera with the three *Tomicus* spp. Moreover, another 33 main genera of the fungal communities associated with the three *Tomicus* spp were identified. Among these, 29 genera were from Ascomycota (*Acremonium*, *Alternaria*, *Aspergillus*, *Beauveria*, *Candida*, *Chaetomium*, *Cladosporium*, *Debaryomyces*, *Fusarium*, *Gibberella*, *Humicola*, *Kuraishia*, *Monascus*, *Mycosphaerella*, *Nakazawaea*, *Neostagonospora*, *Neocamarosporium*, *Ogataea*, *Penicillium*, *Pestalotiopsis*, *Phaeomoniella*, *Phaeococcomyces*, *Phaeothecoidea*, *Phaeomycocentrospora*, *Pseudallescheria*, *Rhodotorula*, *Talaromyces*, *Toxicocladosporium*, and *Yamadazyma*), three genera were from Basidiomycota (*Chionosphaera*, *Cryptococcus*, and *Guehomyces*), and one genus (*Mortierella*) was from Zygomycota. The ecological function of these genera and their role in *Tomicus* spp. cannot be inferred based on HTS analysis; however, the ecologies of certain genera to which these fungi belong have been well characterized, and this should allow us to predict the potential functions of these genera or similar genera.

The most abundant OTUs were taxonomically assigned to yeasts and yeast-like fungi, including seven genera (*Candida*, *Debaryomyces*, *Kuraishia*, *Nakazawaea*, *Ogataea*, *Rhodotorula*, and *Yamadazyma*) of ascomycetous fungi (Saccharomycotina) and three genera (*Chionosphaera*, *Cryptococcus*, and *Guehomyces*) of basidiomycetous fungi. Yeasts are common associates of numerous bark and wood-boring beetles [55], such as ambrosia beetles [14], *Dendroctonus* beetles [56,57,58,59,60,61], and *Ips* beetles [56,62,63]. Yeasts and yeast-like fungi might play critical roles in bark beetle development or fecundity as nutritional and digestive symbionts [59,60,64,65], counteracting host-plant defense, modifying tree chemistry or metabolizing toxic terpenoids, and participating in the production by bark beetles of volatile chemicals used for communication [56,62,63,64,66,67,68]. Furthermore, several yeasts are transported by adult beetles phoretically on the exoskeleton [57,61,69].

Another high-abundance genus associated with the pine-shoot beetles was *Fusarium*. *Fusarium* spp. is generally recovered from plants as parasitic or mutualistic symbionts [70], and *Fusarium* enables the degradation of hemicelluloses and/or cellulose [70,71]. The association of *Fusarium* fungi with bark beetles has been widely reported, such as with *O. erosus* [32,72] and ambrosia beetles [14,71,73,74,75]. *Fusarium* belongs to Hypocreales, and our HTS analysis led to the detection of three other genera from Hypocreales: *Acremonium*, *Beauveria*, and *Gibberella* (the sexual form of *Fusarium*). However, we cannot currently identify the exact species associated with the beetles, and the importance of the interactions of these fungi with bark beetles cannot be readily elucidated. We speculate that the fungi potentially play a notable entomopathogenic, nutritional, symbiotic, or phoretic role during the growth and reproduction periods of bark beetles [76,77].

*Alternaria* fungi are widely reported to be extensively distributed and associated with bark beetles, such as *O*. *erosus*, ambrosia beetles (*Xyleborinus saxesenii*), *Ips typographus*, *I. amitinus*, and *Pityogenes chalcographus* [32,78,79]. However, the role of the *Alternaria* associated with these beetles remains unclear. Furthermore, *Alternaria* belongs to Pleosporales, which also includes *Neocamarosporium*, *Neostagonospora*, and *Phaeomycocentrospora*, and these three other genera were also not detected in *T*. *brevipilosus*, *P*. *kesiya*, and Pu’er samples.

Three genera of Eurotiales (*Aspergillus*, *Penicillium*, and *Talaromyces*) were also retrieved here. *Penicillium* is recovered from plants as a saprotroph and can degrade hemicelluloses and/or cellulose [70] and has also been frequently obtained from galleries infected by bark beetles. Moreover, *Aspergillus* fungi are widely distributed and have been reported in ambrosia beetles [32,78,79]. However, whether the fungus species belonging to these genera play any beneficial or harmful role related to bark beetles is unclear.

Lastly, in this study, we sporadically detected several other genera (*Chaetomium*, *Cladosporium*, *Humicola*, *Mycosphaerella*, *Pestalotiopsis*, *Phaeothecoidea*, *Phaeococcomyces*, *Phaeomoniella*, and *Toxicocladosporium*), which reflects the high fungal diversity of *Tomicus* spp. However, a strict association between these fungi and bark beetles has not been demonstrated, and their relationship and interaction remain poorly understood.

## 5. Conclusions

This study has comprehensively characterized, for the first time, the fungal-community composition and fungal diversity associated with three pine-shoot beetles, *T*. *yunnanensis*, *T*. *minor*, and *T*. *brevipilosus*, by using the HTS approach. Greater species richness and higher promiscuousness of fungal communities were detected using culture-independent analyses here than previously reported using culture-dependent methods. Furthermore, our results reconfirmed some of the beetle-fungus associations reported previously. Although the HTS approach is not suitable for identifying fungi to the species level, the genera to which these fungi could be ranked can also provide information essential for elucidating their potential functions in the interactions between the fungi, *Tomicus* spp., and the host plant. Moreover, a study of this type can help us understand the roles of the associated fungi during shifts in ecologies, hosts, locations, and bark beetles, which can serve as a model for the function and evolution of fungal associates in insects as a whole.

The differences in fungal diversity and abundance associated with bark beetles might also indicate that the infestation intensities of the bark beetles differ. However, it is challenging to clarify, through HTS analysis, the dominant role of the differences in fungal abundance and diversity in bark beetle-fungus-host interaction. Does any link exist between the differential aggressiveness and fungus associates of each shoot beetle, particularly in the sympatric circumstance of various beetles coexisting? Additional deep and accurate investigation is required to answer this question.

## Figures and Tables

**Figure 1 microorganisms-08-00015-f001:**
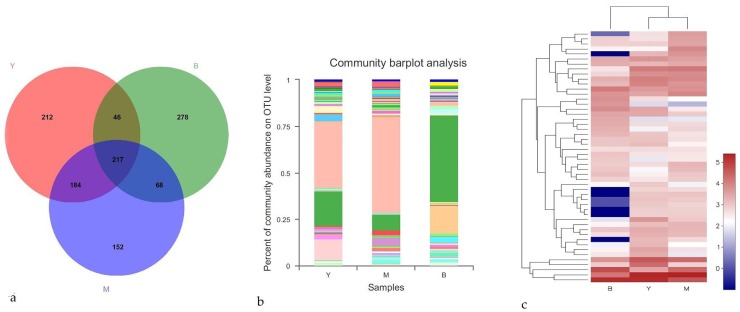
Epibiotic fungal communities associated with *Tomicus* beetles. (**a**) Venn diagram showing shared and unique operational taxonomic units (OTUs) at 97% identity among three *Tomicus* spp.; (**b**) Stack plot showing species richness and evenness of epibiotic fungal communities of the three beetle species. Individual histograms report averages within each species. Each color band represents a distinct fungal OTU; the band width corresponds to the relative abundance of reads of the OTU in the dataset; (**c**) Heatmap of relative abundance of top 50 epibiotic fungal genera of the three beetle species. Upper stacks report summation within each species. Y = *T. yunnanensis*; M = *T. minor*; B = *T. brevipilosus*.

**Figure 2 microorganisms-08-00015-f002:**
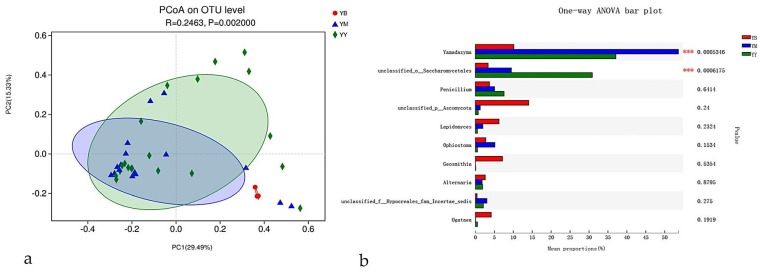
Principal co-ordinates analysis plots of epibiotic fungal communities of *T*. *yunnanensis*, *T*. *minor*, and *T*. *brevipilosus* were collected from *P. yunnanensis*, based on beetle species dissimilarities. Distinct taxa were identified in beetle species groups by using One-way ANOVA analysis; (**a**) Samples labeled according to the three beetle species. (**b**) Extended error-bar plot showing the 10 most abundant genera that exhibited significant differences among the three beetle species. YY = *T. yunnanensis* from *P. yunnanensis*; YM = *T. minor* from *P. yunnanensis*; YB = *T. brevipilosus* from *P. yunnanensis*.

**Figure 3 microorganisms-08-00015-f003:**
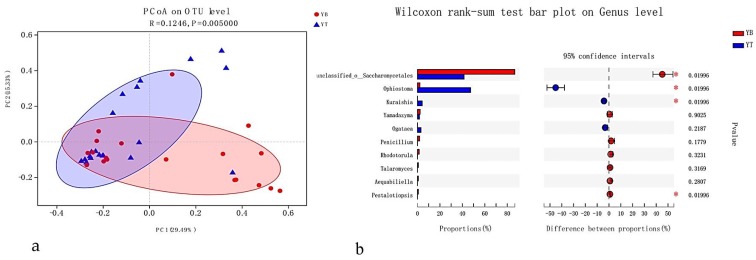
Principal co-ordinates analysis plots of epibiotic fungal communities of *T*. *yunnanensis*, *T*. *minor*, and *T*. *brevipilosus* were collected from *P. yunnanensis*, based on branch/trunk infection section dissimilarities. Distinct taxa were identified in branch/trunk infection section groups by using Wilcoxon rank-sum test analysis. (**a**) Samples labeled according to the infectious sections of the three beetle species; (**b**) extended error-bar plot showing the 10 most abundant genera that exhibited significant differences between branch- and trunk-infection sections. Positive differences in mean relative abundance indicate the genus overrepresented in the branch, whereas negative differences indicate greater abundance in the trunk. YB = Branches infected by *T*. *yunnanensis*, *T*. *minor*, and *T*. *brevipilosus* from *P. yunnanensis*; YT = Trunks infected by *T*. *yunnanensis*, *T*. *minor*, and *T*. *brevipilosus* from *P. yunnanensis*.

**Figure 4 microorganisms-08-00015-f004:**
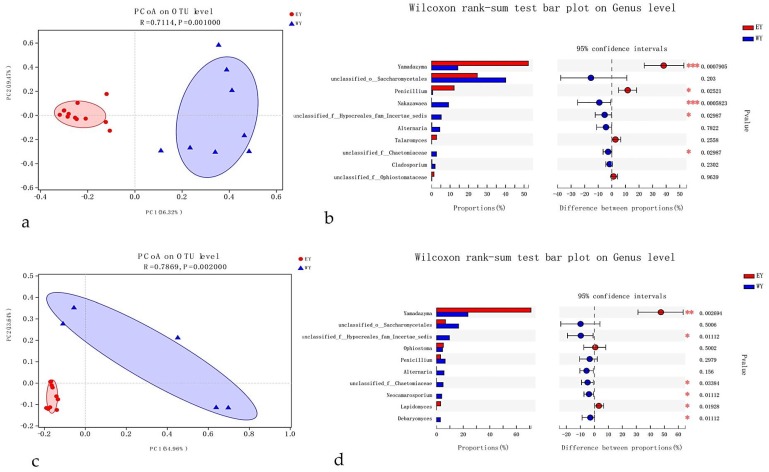
Principal co-ordinates analysis plots of epibiotic fungal communities of *T*. *yunnanensis* and *T*. *minor* were collected from *P. yunnanensis*, based on location dissimilarities. Distinct taxa were identified in distinct beetle locations by using Wilcoxon rank-sum test analysis. (**a**) *Tomicus yunnanensis* samples labeled according to the two locations of the three beetle species; (**b**) extended error-bar plot showing the 10 most abundant genera associated of *T*. *yunnanensis* that exhibited significant differences among the two locations; (**c**) *Tomicus minor* samples labeled according to the two locations of the three beetle species; (**d**) extended error-bar plot showing the 10 most abundant genera associated of *T*. *minor* that exhibited significant differences among the two locations. Positive differences in mean relative abundance indicate the genus overrepresented in Eastern Yunnan, whereas negative differences indicate greater abundance in Western Yunnan. EY = Eastern Yunnan (three sites, Anning, Yuxi, and Qujing near each other, are located around provincial capital Kunming, in eastern Yunnan Province); WY = Western Yunnan (Xiangyun is far away from provincial capital Kunming, in western Yunnan Province).

**Figure 5 microorganisms-08-00015-f005:**
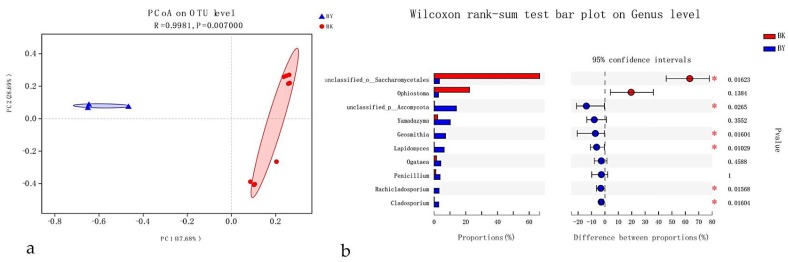
Principal co-ordinate analysis plots of epibiotic fungal communities of *T*. *brevipilosus*, based on host dissimilarities. Distinct taxa were identified in *P*. *yunnanensis* and *P*. *kesiya* groups by using Wilcoxon rank-sum test analysis. (**a**) Samples labeled according to the hosts of *T*. *brevipilosus*; (**b**) extended error-bar plot showing the 10 most abundant genera that exhibited significant differences between *P*. *yunnanensis* and *P*. *kesiya*. Positive differences in mean relative abundance indicate the genus overrepresented in *P*. *kesiya*, whereas negative differences indicate greater abundance in *P*. *yunnanensis*. BK = *T. brevipilosus* of *P. kesiya*; BY = *T. brevipilosus* of *P. yunnanensis*.

**Table 1 microorganisms-08-00015-t001:** Diversity indices for epibiotic fungal communities of the three *Tomicus* beetle species.

Sample\Estimators	Samples Number			Chao 1			Shannon		
	B	T	B-T	B	T	B-T	B	T	B-T
*Tomicus brevipilosus*	7	5	12	118.58 ± 29.43	146.47 ± 33.46	139.9 ± 40.45	1.57 ± 0.97	2 ± 0.82	1.75 ± 0.56
*T*. *minor*	9	7	16	173.84 ± 70.26	64.01 ± 25.91	125.79 ± 77.71	2.12 ± 0.89	1.28 ± 0.41	1.75 ± 0.83
*T*. *yunnanensis*	8	12	20	146.04 ± 69.63	71.6 ± 24.82	101.38 ± 60.07	2.08 ± 1.17	1.31 ± 0.27	1.62 ± 0.86

B = branch; T = trunk; B-T = the SUM/mean (samples number/Chao 1 and Shannon) of branch and trunk.

**Table 2 microorganisms-08-00015-t002:** Ophiostomatoid fungi reported from *Tomicus yunnanensis*, *T*. *minor,* and *T*. *brevipilosus* in the literature.

Taxon	Host	Beetle	References
*Esteya vermicola*	*P*. *yunnanensis*	*Tomicus yunnanensis*	[34]
*Graphilbum anningense*	*P*. *yunnanensis*	*T*. *yunnanensis*, *T*. *minor*	[34]
*Gra*. *fragrans*	*P*. *yunnanensis*	*T*. *minor*	[34]
*Leptographium yunnanensis*	*P*. *yunnanensis*, *P*. *kesiya*	*T*. *yunnanensis*, *T*. *brevipilosus*	[34,44]
*Ophiostoma aggregatum*	*P*. *yunnanensis*	*T*. *yunnanensis*, *T*. *minor*	[34]
*O*. *brevipilosi*	*P*. *kesiya*	*T*. *brevipilosus*	[34,46]
*O*. *canum*	*P*. *yunnanensis*	*T*. *yunnanensis*, *T*. *minor*	[34]
*O*. *ips*	*P*. *yunnanensis*	*T*. *yunnanensis*	[48]
*O*. *minus*	*P*. *yunnanensis*	*T*. *yunnanensis*	[54]
*O*. *quercus*	*P*. *yunnanensis*	*T*. *yunnanensis*	[48]
*Ophiostoma* sp. 1	*P*. *yunnanensis*	*T*. *yunnanensis*	[34]
*O*. *tingens*	*P*. *yunnanensis*	*T*. *yunnanensis*, *T*. *minor*	[34,47]
*Sporothrix abietina*	*P*. *yunnanensis*	*T*. *yunnanensis*	[43]
*S*. *macroconidia*	*P*. *yunnanensis*, *P*. *kesiya*	*T*. *yunnanensis*, *T*. *brevipilosus*	[34]
*S. nebularis*	*P*. *yunnanensis*	*T*. *yunnanensis*	[45]
*S*. *pseudoabietina*	*P*. *yunnanensis*	*T*. *minor*	[34]

**Table 3 microorganisms-08-00015-t003:** Ophiostomatoid fungi OTUs of the three beetle species and their identity as a consensus of searches in the NCBI, GenBank BLASTn and a match with our reference sequenced communities and reference cultures.

Order	Genus	Possible Species	OTU	Reads No.
Ophiostomatales	*Ceratocystiopsis*	*C*. *minima*	OTU379, OTU243	41
	*Graphilbum*	*Gra*. *microcarpum*	OTU434	31
		*Graphilbum anningense*	OTU326	1255
	*Grosmannia*	*Grosmannia aurea*	OTU677	7
		*G*. *yunnanense*	OTU433	527
	*Ophiostoma*	*O*. *macrosporum\* *O. tingens*	OTU391, OTU1178	6349
		*O*. *brevipilosi*	OTU273, OTU272, OTU277, OTU370, OTU278, OTU375, OTU194, OTU281, OTU286, OTU304, OTU306, OTU308, OTU353, OTU332, OTU342, OTU349, OTU348, OTU269, OTU266, OTU263, OTU294, OTU222, OTU221, OTU227, OTU224, OTU223, OTU367, OTU255, OTU59, OTU254, OTU318, OTU317, OTU259, OTU350, OTU354, OTU359, OTU347, OTU210, OTU217, OTU215,	56,779
		*O*. *tapionis*	OTU343, OTU327, OTU248, OTU298, OTU369, OTU360, OTU338, OTU262, OTU203, OTU290, OTU296, OTU1286, OTU393,	27
		*O*. *canum*	OTU393, OTU175	30,863
		*O*. *nigrocarpum*	OTU1286	4
		*O*. *pusillum*	OTU363	37
		*O*. *minus*	OTU963, OTU432	1256
	*Sporothrix*	*Sporothrix* sp.	OTU218	4979

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
