# Peer review of "Epibiotic Fungal Communities of Three Tomicus spp. Infesting Pines in Southwestern China"

_microorganisms, 2019, doi:10.3390/microorganisms8010015_

Round 1

Reviewer 1 Report

Dear authors,

Interesting manuscript you have here. It provides insight into the fungal community of these three beetles.

I think the manuscript is acceptable for publication after a minor revision and some text editing.

Please find your manuscript attached with some comments.

Cheers

Author Response

Review 1:

Point 1. Line 16: Replace "2" for "two". When numbers are less than 10, it is more common to write the numbers in words.

Typically, if you want to use the actual number they must be followed by a unit of measurement.

Response 1: Done as the suggestion. Revised throughout the manuscript.

Point 2. Line 19: When the authors say "periods" do they mean time? Or do they mean sections? Perhaps there is a better word for it.

Point 3. Line182: It is kind of confusing. Does period mean section? or time?

Point 4. Line 230: Infectious period? do you mean December 2016? or March 2017? Please rephrase.

Point 5. Line241-248:Please instead of using the word period, use the word "section" or "infectious points".

Response 2-5: Combined reply to question 2-5. “infection periods” in this article was changed to “infection section”. Revised throughout the manuscript.

Point 6. Line 34: Typically, you don't want to use words that are present in the title. Keywords are used to increase the visibility of your research. Please change these two.

Response 6: Revised according to the suggestion. Delete “epibiotic fungal community; Tomicus spp.” and add three new key words “Tomicus yunnanensis; T. minor; T. brevipilosus

Point 7. Line 52-53: It would be much better if the authors give specific examples (scientific names) rather than just the genera. The genera include several species, several of them are neutral.

Response 7: Revised according to the suggestion. “Dendroctonus valens LeConte, Ips typographus Linnaeus, Scolytus multistriatus Marsham, and T. yunnanensis” was changed to the “genera”.

Point 8. Line 84: delete “of”.

Response 8: Done as the suggestion.

Point 9. Line 97-98: The way is written it seems the authors are extracting beetles DNA. Is this really what you are trying to say? I would like to think that the authors are "washing" the beetles and the aliquots are used for the analysis. Please rephrase. What about something like "due to low concentration of epibiotic fungal DNA of a single beetle, 25-30 beetles were used as a sample"

Response 9: Revised according to the suggestion. It was revised in the manuscript as: “Due to the low concentration of epibiotic fungal DNA of a single beetle, 25-30 beetles were pooled as a sample, depending on the pre-experimental results and number of beetles collected.”

Point 10. Line178-180: It would be ideal if the authors say that diversity was lower in T. brevipilosus than T. yunnanensis and T. minor at the branch infection section.

Response 10: Revised according to the suggestion. It was revised in the manuscript as: “The diversity and richness was lower for T. brevipilosus samples than for T. yunnanensis and T. minor samples at the branch infection section but significant differences were not detected among three beetle species.”

Point 11. Line 234: “species”instead of “variety”.

Response 11: Done as the suggestion. I have revised throughout the manuscript.

Point 12. Line 365: delete unnecessary words.

Response 12: Done as the suggestion.

Point 13. Line 371-372:I would like the authors to explain a little bit more (in a couple of sentences or an example) how do they think the reproductive biology of these beetles (they all belong to the same genera) could explain the difference in the fungal community between T. yunnanennsis and T. minor vs. T. brevipilosus. Certainly host could be a potential driver for such differences but reproductive biology? i would like to see the authors opinion.

Response 13: The reproductive biology of T. yunnanennsis and T. minor vs. T. brevipilosus was different. It has detailed description in the article of Chen et al. (2015) [23]: Tomicus brevipilosus exhibited two general patterns of infestation. From early March to mid-April, T. brevipilosus bred preferentially in the trunks of Yunnan pine trees that were already infested by T. yunnanensis and T. minor, colonizing spaces along the trunk that were not already occupied by the other two Tomicus species. Later, from about mid-April to early June, when there were no Yunnan pine trees newly infested by T. yunnanensis and T. minor, T. brevipilosus attacked Yunnan pine by itself, infesting the lower parts of the trunk first and then infesting progressively upward along the trunk into the crown.

So, it was revised in the manuscript: “Another underlying factor could be the divergent reproductive biology and pattern of infection of different beetles. The breeding attack timing of T. brevipilosus is obviously later than those of other two beetles, with two infection patterns (following the attacks by T. yunnanensis and T. minor in same host, or attacking the host by itself) [23-24]. The period hosts were infected and their healthy status could influence the fungal communities associated with bark beetles [42]. So, the later breeding attack and the different infection patterns comparing to T. yunnanennsis and T. minor could lead to the difference of the fungal community associated with T. brevipilosus.”

Chen, P.; Lu, J.; Haack, R.A.; et al. Attack Pattern and Reproductive Ecology of Tomicus brevipilosus (Coleoptera: Curculionidae) on Pinus yunnanensis in Southwestern China. Journal of Insect Science. 2015, 15(1).

Point 14. Line414-415:Is sampling period the time for beetle collection? is sampling point the part of the tree from which beetles were collected? or sampling point is the location? Anning, Yuxi and/or Qujing?

Response 14: It was revised in the manuscript: “Another point to consider is that sampling time and sampling trees were artificially defined”.

Point 15. Line 419-420:beneficial fungi? to whom, the beetle? How do the authors know the fungi were beneficial and not neutral? What experiments did they carry?

Most of bark and ambrosia beetle symbiosis states that several fungi in the beetles repertoire have unknown functions, some of them are assumed to be nutritional (primary symbionts) but with the rest, there is not 100% certainty.

Response 15: Revised according to the suggestion. It was revised in the manuscript: “which indicates that the fungi carried by bark beetles are distinct and that bark beetles likely carry some neutral or beneficial fungi preferentially under certain conditions depending on their own requirements”.

Point 16. Line 448:This word “insignificantly” is very subjective. Please delete

Response 16: Done as the suggestion.

Point 17. Line 495:this word “Penicillium” is not needed

Response 17: Done as the suggestion.

Point 18. Line 497:However

Response 18: Done as the suggestion.

Reviewer 2 Report

This manuscript provides baseline data on fungal communities of Tomicus spp. infesting pines in Southwest China. While I am not very familiar with some of the methods employed, I find the manuscript to be well written and constructed, and to provide a comprehensive characterization of fungal communities associated with Tomicus spp. within the region.

I only have a few minor suggestions for improvement:

1) Ln. 42: Delete first comma.

2) Ln. 56: Further define what is meant by “greening” coniferous species?

3) Ln. 78: Technically, these are not “varieties”. Revise to “…specific to each bark beetle and host…” Please review and revise throughout the manuscript.

4) The figures need to be enlarged.

5) Ln. 365: Delete “substantially”

6) Ln. 464: Delete “alone”

Author Response

Review 2:

Point 1. Line 42: Delete first comma.

Response 1: Done as the suggestion.

Point 2. Line 56: Further define what is meant by “greening” coniferous species?

Response 2: Revised according to the suggestion. “greening” in this article was changed to “afforestation”.

Point 3. Line 78: Technically, these are not “varieties”. Revise to “…specific to each bark beetle and host…” Please review and revise throughout the manuscript.

Response 3: Combined reply to Review 1 of point 11, “species”instead of “variety”. It was revised in the manuscript: “specific to beetle species”.

Point 4. The figures need to be enlarged.

Response 4: Done as the suggestion.

Point 5. Line 365: Delete “substantially”.

Response 5: Done as the suggestion.

Point 6. Line 464: Delete “alone”.

Response 6: Done as the suggestion.